# *Aspergillus* spp. Isolated from Lungs of Poultry (*Gallus gallus*) at the Mycology Laboratory, School of Veterinary Medicine, Universidad Nacional, Heredia, Costa Rica between 2008 and 2021 and Associated Factors

**DOI:** 10.3390/jof9010058

**Published:** 2022-12-30

**Authors:** Oscar Ulloa-Avellán, Alejandra Calderón-Hernández, Randall Rubí-Chacón, Bernardo Vargas-Leitón

**Affiliations:** 1Mycology Laboratory, School of Veterinary Medicine, Universidad Nacional, Ulloa, Heredia 40104, Costa Rica; 2Integrated Program of Population Medicine, School of Veterinary Medicine, Universidad Nacional, Ulloa, Heredia 40104, Costa Rica

**Keywords:** *Aspergillus*, *Aspergillus* section *Fumigati*, *Aspergillus* section *Flavi*, *Aspergillus* section *Nigri*, *Aspergillus* section *Terrei*, avian aspergillosis, Costa Rica, prevalence, *Paecilomyces variotti*, public health

## Abstract

Aspergillosis is a disease caused by some species of the fungus *Aspergillus*, occurring in both mammals (including humans) and birds, the latter being the most susceptible group. *Aspergillus* must be considered a public health concern as it affects the poultry industry economically and is an occupational risk to its workers. A retrospective study of fungal isolates from the lungs of chickens (*Gallus gallus*), analyzed between 2008 and 2021 at the Mycology Laboratory, School of Veterinary Medicine, Universidad Nacional, Heredia, Costa Rica was performed to report the prevalence of *Aspergillus* spp. in poultry farms in Costa Rica and their associated factors. A total of 1113 cases were received, of which 31% (*n* = 392; 95% CI: 28.3–33.7) were positive for fungal isolation. *Aspergillus* was the most frequently detected genus, and the most frequent sections were *Fumigati* (*n* = 197/392, 50.3%), *Flavi* (*n* = 90/392, 22.9%), and *Nigri* (*n* = 50/392, 12.7%). Significant effects (*p* < 0.05) related to the year, geographical origin, purpose, and age were identified in relation to the *Aspergillus* infection. The identified factors are explained by climatic variations in the tropics and the particularities of the birds. Future research including molecular characterization and antifungal susceptibility tests in animals, humans, and the environment, are needed to better understand the risks of the diseases caused by those fungi in this country.

## 1. Introduction

*Aspergillus* is a genus of filamentous saprophytic fungi, which is found in various environmental substrates such as soil, grains, seeds, as well as in decomposing vegetation [1]; it has a worldwide distribution [2,3]. This genus is composed of six subgenus, 27 sections, 86 series, and more than 695 species [4]. The vast majority inhabit the environment and do not represent a risk to public health, except for species from the sections *Fumigati, Flavi, Nigri, Circumdati, Terrei, Nidulantes, Ornati, Warcupi, Candidi, Restricti, Usti, Flavipedes*, and *Versicolores*, which have been described as agents of disease [5,6].

Aspergillosis is the disease caused by some species of this genus [3,7] that occur in birds and mammals (including humans) [5,7], in which infection happens by the inhalation or ingestion of conidia (asexual reproductive structures) from the environment or, in the case of eggs, by penetration of the shell [2,3]. Inhalation is the main route of entry; thus, respiratory symptoms are the most common symptoms in all affected species [5].

From all over the animal kingdom, avian species are the most susceptible group, including commercial birds such as chickens, turkeys, ducks, geese, ostriches, rheas, quails, and pigeons [8], as well as captive and free-ranging wild species [2,7]. This susceptibility is due to their anatomic particularities such as lack of epiglottis and having a limited number of ciliated cells in the respiratory epithelium, which allows the conidia of these fungi to reach the lower respiratory tract directly; furthermore, not having a diaphragm prevents the active expulsion of particles out of the lungs. At the cellular level, there are also differences compared with mammals, such as the lack of superficial macrophages and having heterophils instead of neutrophils, which make the immune response of birds less effective against these fungi [9].

Avian aspergillosis can occur acutely, mainly in young birds or chronically in adult birds [3,8,10]. *Aspergillus fumigatus* is the principal etiological agent that is responsible for more than 90% of the cases of mortality related to this disease and is reported in approximately 95% of the cases [6].

Infections due to *Aspergillus* spp. are an important cause of mortality related with respiratory problems in birds [1,11], and it is one of the most frequent causes of death of birds in captivity [7]; for example, in captive Magellanic penguins (*Spheniscus magellanicus*), a mortality of 50% was reported [6]. In young birds from 3 days to 20 weeks, outbreaks with high morbidity and mortality can occur, with the mortality rate varying from 4.5% to 90% [6,8].

The economic impact of aspergillosis in the poultry industry is not only related to the mortality, morbidity, low feed conversion, and decreased growth rate of affected birds [2,8,10]; it is also observed in the slaughterhouse, where airsacculitis is one of the main reasons for the condemnation of carcasses [8]. The best example of the economic impact of this disease is the turkey industry from United States, for which annual losses related to turkey mortality from aspergillosis were estimated at 11 million [12].

Being able to infect humans, *Aspergillus* not only economically affects the poultry industry but also poses a risk to workers associated with the industry and public health [12,13]. According to the Global Action Fund for Fungal Infections (GAFFI, Geneva, Switzerland), more than four million people a year are directly affected by infections of this group, which causes more than one million deaths yearly [14]. In addition, multiple cases of antifungal-resistant avian aspergillosis have been reported [7]. In captive centers, azoles are used therapeutically and prophylactically; in the poultry industry, this practice is not common, but antifungals are used to disinfect the environment and bedding [6]. The problem becomes relevant considering the limited availability of drugs to treat infections caused by fungi and the resistance to azoles of *A. fumigatus sensu stricto*, a species commonly related to human aspergillosis [6].

The reported prevalence of *Aspergillus* in birds in Latin America is 32.8% compared with 4.2% in North America. The high prevalence in Latin America may be because tropical areas have more favorable environmental conditions for the growth and proliferation of this fungal agent in the environment [15]. Due to this high prevalence reported in the area and the problems associated with this fungus, it is imperative to generate more information on the situation at the regional level and in each country. To date, there have been no epidemiological or epizootics reports on *Aspergillus* spp. prevalence in human or animal medicine in Costa Rica.

The aim of this research is to report the prevalence of *Aspergillus* spp. in lung samples from poultry of the species *Gallus gallus* in a tropical country in Central America, as well as to find factors that are associated with the frequency of isolation of these fungi in such organs.

## 2. Materials and Methods

### 2.1. Study Design

A retrospective study was carried out from 7 March 2008 to 25 November 2021, considering all mycological cultures of poultry lungs of the species *Gallus gallus*, which were processed and analyzed at the Mycology Laboratory (LMIC), School of Veterinary Medicine, Universidad Nacional, Heredia, Costa Rica. The cases correspond of cultures constructed from control groups of animals before the introduction to the farms or from poultry farms for the diagnosis of aspergillosis sent directly to the LMIC and the Avian Pathology Diagnostic Unit (UDPA) of the same institution.

### 2.2. Sample and Culture Processing

The lungs were directly extracted from the coelomic cavity using dissection scissors and tweezers; later, they were placed in sterile plastic Petri dishes, making pools of up to five pair of organs per plate. They were transported to be immediately cultured or cultured for a maximum of 12 h; in the latter case, the samples were kept refrigerated between 2 and 8 °C. The dissection was performed in the UPDA on a stainless-steel necropsy table accompanied by a Bunsen burner to create a sterile area. The lungs were extracted with forceps and dissecting scissors that were flamed between each bird. At the LMIC, in a type II laminar flow chamber (Labogard, Nuaire, Plymouth, MN, USA), the lungs were cut into portions spanning 1 cm wide at the maximum, then they were passed through the flame three times and cultivated, keeping at least 2 cm of distance between each portion, up to a maximum of seven portions per plate. The samples were cultivated in Sabouraud dextrose agar (Oxoid, Basingstoke, UK; Liofilchem, Roseto Degli Abruzzi, Italy) and were incubated at 37 °C for five days. Once the incubation time had passed, each plate was analyzed for the growth of filamentous hyaline fungi on the surface of the tissue portion; any growth outside this area was considered contamination. The colony-forming units (CFU) were counted and registered in each culture; a case was considered positive when there was growth of at least one CFU of fungus, and a positive case could have more than one type of fungus (mixed isolates).

### 2.3. Fungal Identification

Macroscopic and microscopic morphological identification was carried out following the available guidelines, depending on the year in which the samples were received [4,16,17,18,19,20,21,22]. The macroscopic characteristics that led to the identification of the *Aspergillus* genus included hyaline colonies, circumscribed on the reverse, sometimes with yellowish to reddish pigmentation. On the obverse, they were hyaline colonies ranging from cottony to velvety, which varied in color depending on the conidial pigmentation: blue-green, olive-green, dark brown, black, cinnamon, yellow, and white. Regarding the evaluation of the microscopic morphology, the hyphae were hyaline septate, the conidiophore was rough or smooth, the shape of the vesicle varied from round to semi-clavated, the type of phialide disposition were uniseriate and/or biseriate, and the arrangement of the phialides on the vesicle were radiate, semi-radiate, or columnar. Sexual reproduction structures such as cleistothecia or other structures such as Hülle cells were also considered for the identification. Because the molecular identification was not performed on the isolates, they were classified in the following sections: *Fumigati, Flavi, Nigri, Terrei, Circumdati*, and *Candidi.* In the cases that fungi other than *Aspergillus* were isolated, only those whose growth was pure and directly came from the tissue surface were considered important, and the identification was made following the same references.

### 2.4. Registration of Information and Statistical Analysis

A database was created using Excel software (Microsoft Corporation, Redmond, WA, USA, version 18.2205.10910) with the information from the receipt sheets for each sample, considering the following data: case number of the sample, year of receipt, month it was received, purpose of the bird (incubator, broiler, layer, or reproduction), geographical area of remission such as province (San José, Alajuela, Cartago, Heredia, and Puntarenas) and county of origin, whether or not it came from the airport (if so, that means that the bird lot came from another country), breed, age (in days), sex, and isolated fungus (genus if possible or section in the case of *Aspergillus*), with their respective CFU counts (Appendix A).

Regarding the CFU counts, a table of frequency was created in Excel, and the results were displayed on a heat map created using KNIME (Konstanz Information Miner: Version 4.7.0). Considering that the National Animal Health Service (SENASA) of Costa Rica requires that in cases where at least one CFU of *Aspergillus* is isolated, corrective actions must be performed in the poultry farm; however, no further analysis was performed with that information.

A logistic regression analysis was carried out using the SAS/STAT^®^ program to evaluate the effect of the following factors: year, month, county of origin, purpose, breed, sex, and age of the referred cases, on the occurrence of *Aspergillus* infection. In some cases, due to the low number of samples in certain categories, they were grouped as “others”. The same analysis was specifically performed for the three most frequently isolated sections of *Aspergillus*: *Fumigati, Flavi*, and *Nigri*.

### 2.5. Study Limitations

Because the laboratories in which the birds and the samples were received are in the facilities of a university campus, the reception is not carried out during the holiday periods, which includes Holy Week, two weeks in July, the month of December, and one week in January, impacting the number of samples received in those months. Likewise, due to laboratory opening hours, on some occasions the farms delivered the chick from the incubator on day two as they did not have time to deliver it on day one; for this reason, the incubation area goes from 0 to 2 days, although on farms it is up to one day old. Furthermore, the data collected were the information that was filled out by the senders in the receipt sheet; therefore, on several occasions it is not complete. The identification of the genus *Aspergillus* prior to 2016 was not performed by sections, so in cases where it could not be categorized into any of the most common sections, the isolated fungi was reported as *Aspergillus* spp.

## 3. Results

### 3.1. Prevalence

During the analyzed period, a total of 1113 cases were received, of which 31% (345, 95% CI, 28.3–33.7) were positive for at least one fungus. *Aspergillus* was the most frequently isolated genus, and the most frequent sections of this genus were *Fumigati* (*n* = 197/392, 50.3%), *Flavi* (*n* = 90/392, 22.9%), and *Nigri* (*n* = 50/392, 12.7%). Non-sporulating hyaline fungi and *Paecilomyces variotii* were also detected in the samples (Figure 1).

### 3.2. Colony-Forming Units (CFU)

The count of colony-forming units (CFU) of the isolates showed that most of the fungi growth occurred as a single isolate (*n* = 216); however, in some cases, *Aspergillus* section *Fumigati* (*n* = 3) and *Aspergillus* section *Flavi* (*n* = 2) growth up exceeded 10 CFUs up to a maximum of 24 CFUs (Figure 2).

### 3.3. Logistic Regression Analysis of Aspergillus spp.

The logistic regression analysis showed significant effects (*p* < 0.05) of the year of reception, county of origin, purpose, and age of the birds on *Aspergillus* infection. The categories that pose a greater risk were cases received in 2014 (OR 3.48; IC 95% 1.64–7.37), broilers (OR 2.07; IC 95% 1.18–3.63), poultry intended for breeding (OR 2.94; IC 95% 1.60–5.41), chickens from the county of Puntarenas (OR 10.61; IC 95% 3.90–28.88), and poultry older than 15 days (OR 2.96; IC 95% 1.35–6.50) (Figure 3). There were no significant differences about the months when the samples were submitted or the sex of the chicken.

Regarding the breed significant differences, they were noted in “others” (OR 3.14; IC 95% 1.31–7.56) and for poultry in which the breed was not specified (OR 1.79; IC 95% 1.06–3.02) (Figure 4).

### 3.4. Mixed Isolates

The results indicate that 11.7% (*n* = 45) of the isolates were mixed, meaning that they involved two or three different types of fungi. The combination of *Fumigati* + *Flavi* sections was the most common with 2.25% (95% CI 1.37–3.11, *n* = 25), followed by *Fumigati* + *Nigri* with 1.26% (95% CI 0.60–1.91, *n* = 14), and *Fumigati* + *Terrei* with 0.27% (95% CI −0.03–0.57, *n* = 3); the combinations of *Fumigati* + *Flavi* + *Nigri* and *Nigri* + *Terrei* were found with the same frequency of 0.18% (*n* = 2), and *Flavi* + *Candidi, Flavi* + *Aspergillus* spp. and *Terrei* + *Aspergillus* spp. had a frequency of 0.09% (*n* = 1).

### 3.5. Isolations from Bird Lots Coming from the International Airport

During the study period, a total of 129 samples from bird lots were received from other countries, of which 46 positive cases (35.7%) were obtained without finding a significant difference in relation to chickens from Costa Rica. The *Fumigati* section was the one that was most frequently found (*n* = 37), and the *Flavi, Nigri* and *Terrei* sections were found in equal proportion (*n* = 5 each).

### 3.6. Aspergillus Section Fumigati

Regarding the isolates of *Aspergillus* section *Fumigati*, a significant effect was only found in the year 2014 (OR 2.59; 95% CI 1.17–5.77) and in “other” breeds (OR 5.57; 95% CI 2.30–13.50) (Figure 5).

### 3.7. Aspergillus Section Flavi

In relation to the positive isolates of *Aspergillus* section *Flavi*, significant differences were found in the years 2012 (OR 9.20; 95% CI 2.27–37.40), 2013 (OR 7.75; 95% CI 1.58–38.02), 2014 (OR 13.75; 95% CI 3.41–55.39), 2016 (OR 5.73; 95% CI 1.17–28.01), 2017 (OR 6.26; 95% CI 1.24–31.63), and 2019 (OR 6.62; 95% CI 1.46–29.96). In addition, there were significant differences found in broilers (OR 7.06; 95% CI 2.91–17.13), chickens without indicated purpose (OR 7.41; 95% CI 2.39–22.94), and chickens older than 15 days of age (OR 7.06; 95% CI 2.91–17.13) (Figure 6).

### 3.8. Aspergillus Section Nigri

Regarding the positive isolates of *Aspergillus* section *Nigri*, significant differences were observed in chickens intended for breeding (OR 5.46; 95% CI 1.33–22.34), those from the county of Puntarenas (OR 15.49; 95% CI 2.11–113.96), and the Isa Brown breed (OR 5.045; 95% CI 1.42–17.80) (Figure 7).

## 4. Discussion

Reports on the prevalence of *Aspergillus* in bird lungs are scarce, even though it is a causative agent of an important disease in these animals. The available research varied in terms of the geographic location (the majority are in the tropic), types of samples taken (including bedding, environment surfaces of the enclosures, and food, as well as in various organs such as the lungs, air sacs, brain, gizzard, and liver), bird species considered (productive and wild), and diagnostic methodology (clinical and/or laboratory).

The prevalence obtained in Costa Rica is lower than the prevalence reported in Egypt [23], where it was obtained that 74% of *Aspergillus* isolates that came from animals had signs suggestive of aspergillosis. A review of common respiratory diseases in poultry carried out in Bangladesh mentioned that 6.14% of 1981 birds with respiratory signs were diagnosed with aspergillosis, but only by clinical findings without laboratory confirmation [24]. More recently and also in Bangladesh, a 36% (18/50) prevalence of this fungus was reported in chicken lung samples, confirmed by culture and polymerase chain reaction (PCR) analyses [25].

The occurrence of *Aspergillus* in environmental samples from poultry farms in the Apulia region of southern Italy ranged from 31.6 to 55.1% depending on whether it came from air samples (30/57), feces pools (38/69), or feeders (6/19) [1]. In Egypt, a study was carried out on 100 samples of chicken lungs and 80 samples of feed, bedding, water, and air from farms, obtaining an occurrence of 24% and between 10 and 55%, respectively [26]. Also in Egypt, *Aspergillus* was isolated from a sampling carried out in 88 broilers at a prevalence of 43.2% (38/88) [27]. In that same country, laryngeal swabs were taken from poultry, and the obtained positivity for this fungus was 72.5% (29/40) [28].

In a study carried out in Indonesia with lungs from chickens offered in the market, 66.7% (20/30) were positive for *Aspergillus* [29]. In a study from Iran conducted on 600 broilers, *Aspergillus* was isolated from tracheal swab samples in 51.6% (310/600) of the samples [30]. In Mexico, a sampling was carried out in 73 birds with clinical signs of aspergillosis, and an occurrence of 9.6% of *A. fumigatus* was obtained [31].

In most of the studies consulted, *A. fumigatus* was the most prevalent, ranging from 9.6 to 34.5% [1,23,25,26,27,31]; the findings obtained in Costa Rica confirmed the same. However, in investigations carried out in Iran and Indonesia, *Aspergillus niger* was the most isolated, having values of 37.5% (225/600) and 40% (12/30), respectively [29,30]. In another investigation carried out in Egypt, *Aspergillus flavus* was the most common, at 60% (24/40) [28]. Mixed isolates have also been obtained in other investigations in low proportions [25,28,30]. Regarding the occurrence of *Aspergillus terreus, Aspergillus candidus*, and *Aspergillus ochraceus* (belonging to *Aspergillus* section *Circumdati*), it has also been reported with less frequency [23,31].

*Paecilomyces variotii* has not been associated with disease in poultry, but in human medicine, there are some reports of pneumonia, mainly in immunocompromised patients [32]. Considering its thermotolerance and presence in the dust of poultry farms [33], we highlight this finding to be followed for animal health and occupational risk purposes.

Fungi of the genus *Aspergillus* more easily reproduce when environmental conditions are favorable, especially in warm and humid climates [34]. In the present research, a positive effect of *Aspergillus* infection was detected in the year 2014. During that year, at the Sabana Larga Pluviometry Station located in Atenas, Alajuela, Costa Rica (one of the geographical areas where most of the poultry farms in this country are concentrated), a record of very low rainfall was recorded in July, with an anomaly of rainfall of −85.7mm compared with the historical average of 1940–2008. Moreover, the temperatures in the province of Alajuela were higher than the historical average reported [35]. Added to this, in the first half of 2014 and 2015, lower than normal temperatures were reported in the tropical sector of the Atlantic Ocean and the Caribbean Sea, and this cooling caused variations in the climate, significantly influencing the rainfall on the Pacific slope. In addition, in the second semester of both years, there was an increase in significant rainfall, and the effect was greater in 2014 due to the warming of the Atlantic waters and the low intensity of El Niño; in 2015, this increase was overshadowed by the strong intensity of El Niño [36].

*Aspergillus* infections have been documented in all ages of birds, but the extremes of age are the most affected [37]. In the present study, a higher risk was obtained in birds older than 16 days, such as in the research in Bangladesh [25], in which a higher infection of *Aspergillus flavus* was reported in animals older than 3 weeks. In addition, also in Bangladesh, a higher mortality was observed and associated with aspergillosis in chickens of Isa Brown and Hy-Line breeds from three to eight weeks of age [34].

Regarding the productive purpose of the birds, in the cases obtained in Costa Rica, a higher risk was observed in animals destined to reproduction and broilers, contrary to the Nigerian research made in clinically sick chickens, where the most affected were the layers and where serious economic losses occur in the growth stage and just before the start of production [38]. Regarding broilers, in a study which compared the effect of an experimental infection by *A. fumigatus* in a line of White Leghorns (layers) and a line of broilers, it was observed that the layers did not present mortality compared with the broilers, which presented a mortality greater than 50%. Furthermore, it was shown that the layers were able to resolve the infection more easily than the broilers [39]. This may be due to the immunological differences of each line of production as layers show two types of well-developed immune response (antigenic and cellular) as opposed to broilers, which only show a developed antigen-specific immune response [40].

In reference to a greater risk detected in the geographical area of Puntarenas, Costa Rica, a possible explanation could be that it is the only coastal province where samples were submitted, so the salinity, humidity, and temperature conditions are different from the greater metropolitan area (located in the central valley of the country). At the same time, the greatest risk of isolating section *Nigri* in this province could be attributed to this fungus halotolerance [21,22], which coincides with the study carried out in Payakumbuh, Indonesia (a coastal zone) in broiler chicken lungs, where *A. niger* was most frequently isolated [29]. Furthermore, this fungus was the second most isolated section in samples of laryngeal swabs from poultry and the most frequently isolated in sputum samples of poultry workers in Quena, Egypt [29]; in Bangladesh, this fungus was found in higher numbers in samples from 2–3-week-old birds [25].

Detection of *Aspergillus* in the lungs, environment, surface, food, and litter of birds has implications for public health, the occupational health of farmworkers, and animal health and welfare [1,33,37,38]. Aspergillosis is a sapronotic disease, which means that the infection is acquired by inhalation or ingestion of conidia from a contaminated environment, which can cause respiratory signs in immunocompromised patients and an asthma-like allergic reaction in people exposed to high concentrations or people who are repeatedly exposed to this fungus [22,41]. Poultry farm personnel are an occupational risk group due to the continuous exposure to contaminated environments [1,33,37,38]; the dust from avian farms contains a combination of microorganisms and organic materials from the animals [42]. Biosecurity measures on farms are aimed at avoiding contagion between birds due to movement of personnel, but they must also include the use of personal protective equipment for the workers at risk of environmental exposure.

According to the present study, future research must be intended to identify those fungi at the species level using PCR and sequencing analyses to survey the presence of cryptic species that could be more virulent or refractory to treatment. Moreover, it must involve antifungal susceptibility tests to create a perspective of the antifungal resistance of the isolates, including samples from animals, humans, food, and the environment. All this information will improve the understanding of the epizootiology and the epidemiology of the diseases caused by them in this country, and it will contribute to minimizing the loses of the poultry industry while supporting public health policies with a One Health background.

Mycotic diseases (both from human and animal medicine) are prevalent in tropical areas; however, many of the countries located in that geographical zone are under developing economies, and the reports of fungal agents and diseases are underestimated and/or lacking molecular methods to provide robust information for diagnosis and epidemiology. Furthermore, the difficulty in standardizing and acquiring antifungal susceptibility tests or the availability of less harmful antifungal treatments enhances the importance of funding opportunities and international collaboration to train and integrate professionals of those countries to close the gap regarding diagnosis and research in those populations.

## Figures and Tables

**Figure 1 jof-09-00058-f001:**
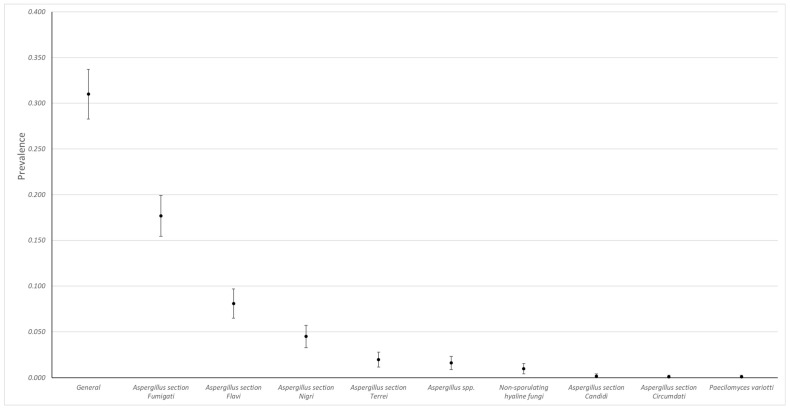
Observed prevalences (95% CI) of fungi isolated in chicken lungs (*Gallus gallus*) ranging from 2008 to 2021 in the Mycology Laboratory, School of Veterinary Medicine, Universidad Nacional, Heredia, Costa Rica.

**Figure 2 jof-09-00058-f002:**
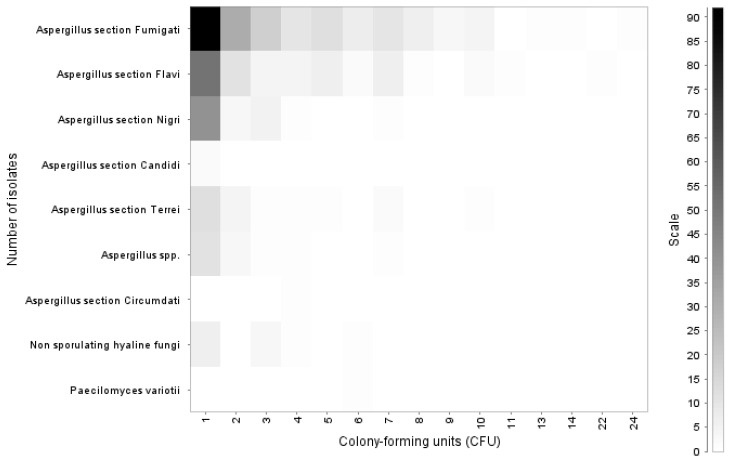
Colony-forming units of fungi isolated in chicken lungs (*Gallus gallus*) during 2008–2021 in the Mycology Laboratory, School of Veterinary Medicine, Universidad Nacional, Heredia, Costa Rica.

**Figure 3 jof-09-00058-f003:**
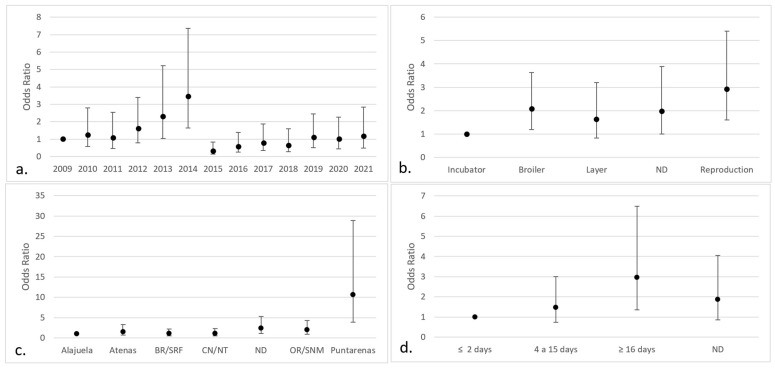
Categories with the highest risk of positive fungal isolates in chicken lungs (*Gallus gallus*) from 2008 to 2021 in the Mycology Laboratory, School of Veterinary Medicine, Universidad Nacional, Heredia, Costa Rica. (**a**) Year of remission; (**b**) productive purpose of the bird; (**c**) location (county) of origin; (**d**) bird age. 2009 includes 2008; ND = non determinate productive purpose; BR/SRF = Barva and San Rafael; CN/NT = counties from the center and north of Alajuela; ND = non-determinate origin and Dota, Santa Ana, Sarapiquí, Turrialba, Turrubares, Upala, Río Cuarto, San Carlos, and Pavas; OR/SNM = Orotina and San Mateo; ND = non-determinate age of the birds.

**Figure 4 jof-09-00058-f004:**
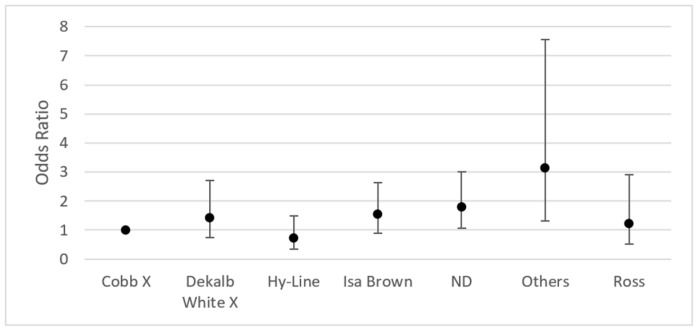
Odds ratio by breed (95% CI) of positive isolates for fungi in chicken lungs (*Gallus gallus*) from 2008 to 2021 in the Mycology Laboratory, School of Veterinary Medicine, Universidad Nacional, Heredia, Costa Rica. Cobb X = pure Cobb mixed with Hubb and Ross; Dekalb White X = pure Dekalb White mixed with Isa Brown; ND = non-determinate breed; Others = Redbro, Sasso, Hubbard, and Lohman.

**Figure 5 jof-09-00058-f005:**
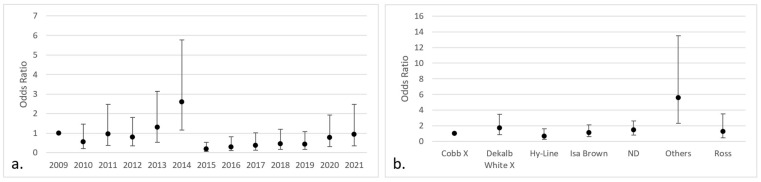
Categories with the highest risk of positive isolates for *Aspergillus* section *Fumigati* (95% CI) in chicken lungs (*Gallus gallus*) from 2008 to 2021 in the Mycology Laboratory, School of Veterinary Medicine, Universidad Nacional, Heredia, Costa Rica. (**a**) Year of remission; (**b**) bird breed. 2009 includes 2008; Cobb X = pure Cobb mixed with Hubb and Ross; Dekalb White X = pure Dekalb White mixed with Isa Brown; ND = non-determinate breed; Others = Redbro, Sasso, Hubbard, and Lohman.

**Figure 6 jof-09-00058-f006:**
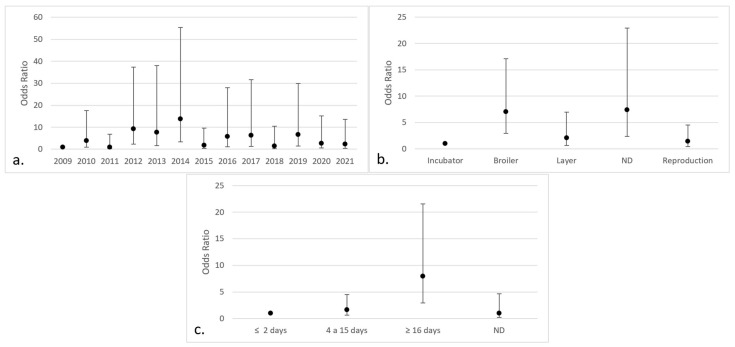
Categories with the highest risk of positive isolates for *Aspergillus* section *Flavi* (95% CI) in chicken lungs (*Gallus gallus*) from 2008 to 2021 in the Mycology Laboratory, School of Veterinary Medicine, Universidad Nacional, Heredia, Costa Rica. (**a**) Year of remission; (**b**) productive purpose of the bird; (**c**) bird age. 2009 includes 2008; ND = non-determinate production purpose or age of the bird.

**Figure 7 jof-09-00058-f007:**
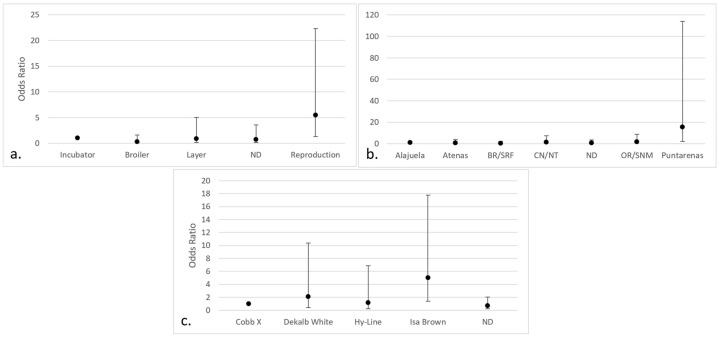
Categories with the highest risk of positive isolates for *Aspergillus* section *Nigri* (95% CI) in chicken lungs (*Gallus gallus*) from 2008 to 2021 in the Mycology Laboratory, School of Veterinary Medicine, Universidad Nacional, Costa Rica. (**a**) Productive purpose of the bird; (**b**) county of origin; (**c**) bird breed. ND = non-determinate production purpose; BR/SRF = Barva and San Rafael; CN/NT = counties from the center and north of Alajuela; ND = non-determinate origin and Dota, Santa Ana, Sarapiquí, Turrialba, Turrubares, Upala, Río Cuarto, San Carlos, and Pavas; OR/SNM = Orotina and San Mateo; Cobb X = pure Cobb mixed with Hubb and Ross; Dekalb White = pure Dekalb White mixed with Isa Brown; ND = non-determinate breed and Ross, Redbro, Sasso, Hubbard, and Lohman.

## Data Availability

The data presented in this study are available in this article and Appendix A.

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
