# Peer review of "Aspergillus spp. Isolated from Lungs of Poultry (Gallus gallus) at the Mycology Laboratory, School of Veterinary Medicine, Universidad Nacional, Heredia, Costa Rica between 2008 and 2021 and Associated Factors"

_jof, 2022, doi:10.3390/jof9010058_

Round 1
Reviewer 1 Report
This study based on the isolation of Aspergillus spp. from lungs of poultry is very pertinent as it has implications not only for the poultry industry itself but also for the workers that directly manage the poultry.
It would benefit from quantitative measures, either from colony counting or from molecular based qPCR analysis. The authors mention the absence of molecular analysis but what about colony counting to have an idea of, not only the type of fungus but also the amount found in each setting? Would it be possible to address that question, having in mind that is a retrospective study? Even if not all samples are represented?
Author Response
Dear Reviewer 1:
Thank you for taking your time to read and make valuable comments to the article. Please see the attachment with the reply of your suggestion.
Point 1: “It would benefit from quantitative measures, either from colony counting or from molecular based qPCR analysis. The authors mention the absence of molecular analysis but what about colony counting to have an idea of, not only the type of fungus but also the amount found in each setting? Would it be possible to address that question, having in mind that is a retrospective study? Even if not all samples are represented?”
Response 1: Since the registers of the colony-forming units (CFU) were recorded in all the cases, I include that information in the database. Then, made a separate table of frequency in Excel and transform it in a heat map using KNIME (Konstanz Information Miner: Version 4.7.0) for presenting the data. However, considering that the National Animal Health Service (SENASA) of Costa Rica request the poultry farms to perform corrective actions with the finding of just one CFU of Aspergillus, no further analyses were made. Following the reviewer suggestion, the CFU information was incorporated in three sections of the methodology and in a new section of results as follows:
Page 3, section 2.2. Sample and culture processing, lines 116-118: “The colony-forming units (CFU) were counted and registered in each culture, a case was considered positive when there was growth of at least one CFU of fungus…”.
Page 3, section 2.4. Registration of information and statistical analysis, lines 146-151: “… with their respective CFU counts (Supplementary Table 1).
Regarding the CFU counts a table of frequency was made in Excel and then results was displayed on a heat map made with KNIME (Konstanz Information Miner: Version 4.7.0). Considering that the National Animal Health Service (SENASA) of Costa Rica requires that in cases where at least one CFU of Aspergillus is isolated corrective actions must be done in the poultry farm, not further analysis was done with that information.”.
Page 5, section 3.2. Colony-forming units (CFU), lines 183-190: “3.2. Colony-forming units (CFU)
The count of colony-forming units (CFU) of the isolates showed that most of the fungi growth as a single isolate (n=214), however in some cases Aspergillus section Fumigati (n=3) and Aspergillus section Flavi (n=2) growth up exceeding 10 CFU until a maximum of 24 CFU (Figure 2).
Figure 2. Colony-forming units of fungi isolated in chicken lungs (Gallus gallus) during 2008-2021 in the Mycology Laboratory, School of Veterinary Medicine, Universidad Nacional, Heredia, Costa Rica.”
The numbers of the next sections of the results and their correspondent figures were changed successively because of the inclusion of this new section and figure. The supplementary table was changed with the one that includes the CFU counts, and the new figure was added to the zipped file of figures.
Thank you for taking the time to read the document and make valuable comments, and I am available to make any further changes for the improvement of this article.
Best regards and happy holidays!

Reviewer 2 Report
The article entitled "Aspergillus spp. isolated from lungs of poultry (Gallus gallus) at the Mycology Laboratory, School of Veterinary Medicine, Universidad Nacional, Heredia, Costa Rica between 2008-2021 and associated factors", deals with a retrospective study of fungal isolates from lungs of chickens, their molecular characterization and antifungal susceptibility tests in animal and human was explained in detail.
From my point of view, the article was well-written and clearly explained all the scientific factors. So I Suggest the article can be published in this journal
Author Response
Thank you for taking the time of reading the document and your comments.
Best regards,